# Detailed Determination of Delamination Parameters in a Multilayer Structure Using Asymmetric Lamb Wave Mode

**DOI:** 10.3390/s25020539

**Published:** 2025-01-18

**Authors:** Olgirdas Tumšys, Lina Draudvilienė, Egidijus Žukauskas

**Affiliations:** Ultrasound Research Institute, Kaunas University of Technology, LT-51423 Kaunas, Lithuania; lina.draudviliene@ktu.lt (L.D.); e.zukauskas@ktu.lt (E.Ž.)

**Keywords:** non-destructive testing, Lamb waves, delamination parametrization, phase velocity dispersion curve, finite element modelling, wind turbine blade, zero-crossing

## Abstract

A signal-processing algorithm for the detailed determination of delamination in multilayer structures is proposed in this work. The algorithm is based on calculating the phase velocity of the Lamb wave A_0_ mode and estimating this velocity dispersion. Both simulation and experimental studies were conducted to validate the proposed technique. The delamination having a diameter of 81 mm on the segment of a wind turbine blade (WTB) was used for verification of the proposed technique. Four cases were used in the simulation study: defect-free, delamination between the first and second layers, delamination between the second and third layers, and defect (hole). The calculated phase velocity variation in the A_0_ mode was used to determine the location and edge coordinates of the delaminations and defects. It has been found that in order to estimate the depth at which the delamination is, it is appropriate to calculate the phase velocity dispersion curves. The difference in the reconstructed phase velocity dispersion curves between the layers simulated at different depths is estimated to be about 60 m/s. The phase velocity values were compared with the delamination of the second and third layers and a hole drilled at the corresponding depth. The obtained simulation results confirmed that the drilled hole can be used as a defect corresponding to delamination. The WTB sample with a drilled hole of 81 mm was used in the experimental study. Using the proposed algorithm, detailed defect parameters were obtained. The results obtained using simulated and experimental signals indicated that the proposed new algorithm is suitable for the determination of delamination parameters in a multilayer structure.

## 1. Introduction

Recently, composite materials have been used in a wide range of structures to reduce weight and increase strength. A composite material is a combination of several materials that have different mechanical, electrical, and thermal properties [1]. Therefore, this type of material has great advantages such as low overall cost, high strength-to-weight ratio, design flexibility, directional properties, and resistance to corrosion, high temperature, and severe fatigue [2,3,4]. However, the combination of several materials leads to inhomogeneity and an anisotropic nature, which poses a serious problem for the effective detection of defects [5,6]. Since, in practice, these advanced composite materials are susceptible to a variety of internal damage [7,8], which can occur during operation or during manufacture, the identification of such defects becomes a complex process [4,9]. It is therefore very important to detect, diagnose, and prognose these damages at an early stage before they propagate to major damage and reach catastrophic failure [8,10]. Delamination is named as one of the most common defects contributing to the degradation of the mechanical properties of these materials, which can lead to the failure of the overall composite component [1]. One of the important aspects of delamination detection in a composite structure is to predict the size of the delamination and at what depth it exists [11]. The delamination can be inspected by techniques that require investigation through the thickness of the structure. Thus, different approaches are required for the estimation of delamination and defects in such types of structures, especially in the case of complex objects under investigation [12]. Non-destructive testing (NDT) and Structural Health Monitoring (SHM) technologies are commonly used for the evaluation of the condition status of complex objects [10,12]. The application of these two technologies generally consists of two parts. First, a measurement system is used to excite and receive waves. Second, data analysis is performed using appropriate signal processing techniques [8,10]. Thus, special techniques are continuously being developed to increase the probability of detecting a fault with a high level of confidence [1]. For the qualitative assessment of multilayer materials, the ultrasonic guided wave-based method [5,8,13,14], acoustic emission (AE) [15], electromechanical impedance (EMI) [16], computed tomography (CT), microwave, thermography, radioscopy, eddy currents, and other techniques are used [12,17,18,19]. Among them, the Ultrasonic Lamb Wave (ULW) method has been extensively investigated regarding the detection of delamination and has been identified as one of the most reliable and promising methods with good results [3,13,14,20,21,22].

Lamb waves are elastic waves created by the motion of particles between the top and bottom surfaces of the plate and propagation between these boundaries. Thus, these waves are more sensitive to various kinds of defects, can travel long distances, have a large detection area, have superior inspection sensitivity, and require minimal equipment [23]. This means that the propagation characteristics of Lamb waves in a specific medium contain information about the current state of the structure, possible defects, and their location. Therefore, various Signal Processing Methods (SPMs) have been developed to extract and parameterize Lamb wave signals, which contain a wide variety of useful information. Thus, data analysis can be implemented through multi-dimensional Fourier transform, Wavelet Transform (WT), Wigner–Ville Distribution (WVD), Hilbert–Huang transform (HHT), and other methods [24,25,26,27,28,29].

However, despite all the good features of Lamb waves, the application of such waves has some limitations due to the unusual propagation characteristics in a specific medium. The phenomenon of dispersion, the infinite number of modes, interference, and the mode conversation are main unusual wave characteristics that make defect and/or delamination detection and parameterization difficult [23,30,31]. Thus, in the general case, the propagation of Lamb waves depends on the elastic properties, density, and thickness of the test material and the frequency of the excited wave and is described by Rayleigh–Lamb equations [5,32] represented by families of dispersion curves. For isotropic plates, this propagation is straightforward and can be described in a simple way. However, if the Lamb waves propagate in a multilayer composite structure, their propagation process becomes more complicated. The propagation of waves depends on the parameters of the individual layers and the interaction between their surfaces [9]. Thus, by knowing the parameters of the individual layers and considering the position of the layers relative to each other, it is possible to calculate the dispersion curves for the propagation velocity of Lamb waves. The calculated velocity of the dispersion curve can thus parameterize the layering.

The aim of this research is to propose a signal-processing algorithm for the detailed determination of delamination parameters in a multilayer structure and to investigate the feasibility of the proposed method for determining delaminations at different depths.

In this paper, Section 2 introduces the methodology for detecting delaminations using Lamb waves, the algorithm for delamination parametrization, and the processing of finite element modelled signals. Section 3 describes the experimental study. Section 4 concludes this paper.

## 2. Methodology for Detecting Delamination’s Using Lamb Waves

### 2.1. Dispersion Theory of Lamb Waves for Composite Structures with Delamination’s

As mentioned in the introduction, Lamb waves are very sensitive to any changes in a structure. Thus, at the point of layering, the structure becomes thinner, which means that the geometry of the structure changes. In this case, if delamination occurs between layers, the propagation of Lamb waves should change. To confirm such a theory, an example is chosen which is a three-layer Glass Fiber-Reinforced Plate (GFRP), where the thickness of the individual layers is *d*_1_ = *d*_2_ = *d*_3_ = 5 mm (Figure 1a). The parameters of the layers of the chosen plate are the same: density *ρ* = 1828 kg/m^3^, Young’s modulus *E*_1_ = 42.5 GPa, Young’s modulus *E*_2_ = 10 GPa, Poisson’s ratio *ν*_12_ = 0.26, Poisson’s ratio *ν*_23_ = 0.4, and in-plane shear modulus *G*_12_ = 4.3 GPa. Two cases of delamination are considered at different depths between layers. In the first case, there is a delamination of width *dl*_1_ between the first and second layers at depth *dd*_1_, and in the second, there is a delamination of width *dl*_2_ between the second and third layers at depth *dd*_2_, as shown in Figure 1a. In order to evaluate the Lamb wave propagation in a multilayer GFRP with delaminations at different depths, the phase velocity dispersion curves were calculated using the semi-analytical finite element (SAFE) method [33]. The phase velocity dispersion curves calculated by the SAFE method in the defect-free region and in both selected delamination regions are shown in Figure 1b.

Analysis of the obtained phase velocity dispersion curves leads to several conclusions. Firstly, it is observed that only the fundamental A_0_ and S_0_ modes propagate in the lower frequency ranges. The cut-off frequency of the higher modes is approximately 50 kHz (Figure 1b). Secondly, the resulting dispersion curves show that changes in geometric thickness in delamination regions affect the phase velocity of the A_0_ mode in the lower frequency region. In contrast, the S_0_ mode is more sensitive to these changes at higher frequencies (Figure 1b). Therefore, for further study, the A_0_ mode at the 50 kHz frequency range is chosen for the study. The higher reduction in the phase velocity of A_0_ mode at the 50 kHz frequency range is obtained when the delamination is between the first and second layers at depth *dd*_1_ (Figure 1b, red line). Meanwhile, a smaller change in phase velocity is observed when analyzing the delamination of width *dl*_2_ between the second and third layers at depth *dd*_2_ (Figure 1b, blue line). Thus, the calculation of the phase velocity dispersion curves could be a parameter for the determination of the depth of layering.

The change in the phase velocity of the A_0_ mode at low frequencies was used by the authors [34] to determine the geometric dimensions (width) of the internal defect of the WTB. Thus, in order to determine, in detail, the parameters of defects and delaminations in a multilayer structure, it is necessary to know not only the area but also the depth at which they are located. A new signal-processing algorithm is therefore developed after using the already described methods [9,33] of digital signal processing, enabling the determination of the location and geometric dimensions of delamination defects. A detailed description of this algorithm is provided in the following subsection.

### 2.2. Algorithm for Delamination Parametrization

A signal-processing algorithm based on the estimation of phase velocity variations in the A_0_ mode is proposed to determine the size, location, and depth of a delamination in a multilayer structure. The flow chart of the proposed signal-processing algorithm is presented in Figure 2.

The algorithm displays the signals emitted by the object in the initial stage in a B-scan image. Filtering is then performed to remove reflections from the delamination edges and to highlight the A_0_ mode. Spatial filtering based on two-dimensional fast Fourier transform (2D-FFT) is used for this purpose [34]. An inverse two-dimensional fast Fourier transform is used to reconstruct a B-scan image that reflects only the propagation of the A_0_ mode. This procedure has been described in detail in a previous paper [34]. Further processing of the received B-scan image is divided into two parts.

In the first part, a defect-sizing algorithm is applied. According to the algorithm presented, the size of the defect (delamination) is determined by measuring the change in phase velocity with respect to the distance with the moving window [34]. The phase velocity is calculated using two adjacent signals acquired at two different spatial measurement positions. In this way, the phase velocity variation is obtained in the defective region that indicates the location of the delamination. Then, the middle threshold between the maximum and minimum values (0.5 or 6 dB level) of the determined phase velocity in the defect region is determined and the distance between two points is designated as the delamination size.

In the second part, the location of the delamination depth is determined. A segment of the phase velocity dispersion curve is used to determine the more accurate depth location. The zero-crossing method is proposed to apply for the phase velocity dispersion curve calculation. This method has been previously proposed and presented in our work [35]. The method involves determining the time instants at which signals cross the zero-amplitude line to calculate phase velocity and frequency. Thus, the method consists of two main parts: obtaining phase velocity values and calculating frequency values. Phase velocity values are calculated using two adjacent signals and the instances at which both signals cross the zero-amplitude line. For the calculation of the frequency values, the determined time instances of the second signals are used. Then, the frequency values are calculated using the duration of each selected half-period.

In order to bring theoretical research closer to real experimental measurements, a fragment of a WTB was chosen as the object of theoretical modelling to verify the effectiveness of the proposed algorithm.

### 2.3. Processing of Finite Element Modelled Signals by the Proposed Algorithm

A WTB is heavily exposed to fluctuating wind or cyclic loads, making it the most potentially defective element of a wind turbine. Therefore, the regular maintenance and inspection of WTBs is necessary. A graphical representation of the multilayer structure of a WTB with delamination in one of the interlayers is presented in Figure 3. The orientations of the main layers of the GFRP in the composite structure are +45°/−45° and 0°/90°/+45°/−45°/0°, respectively. The properties of the materials of the different layers are presented in Table 1. The dispersion curves of the phase velocities of the asymmetric and symmetric modes of Lamb waves were calculated using the SAFE method; the obtained results are presented in Figure 4.

The simulation of Lamb wave propagation in the WTB composite structure was performed using Abaqus finite element software. An explicit algorithm was used to solve the transient wave equation. The structure of the WTB was simplified using a 2D plane strain model. A delamination was modelled by disconnecting relevant surfaces at desired positions. The size of the delamination was 81 mm. The model was meshed with rectangular mapped mesh. The size of the element was 0.25 mm. In order to excite the A_0_ Lamb waves, transient excitation force was applied to the surface of the object under investigation. The width of the excitation zone was 30 mm. For the excitation of guided waves with 43 kHz frequency, five periods sine burst with Gaussian envelope was used. To selectively excite the A_0_ mode, applied force was phased by delaying the force applied to each node by the following law:(1)$$∆t=\frac{sin(\theta )\cdot dx}{c_{A0}}$$
where *θ* is the wave propagation angle, which in our case is 90°; d*x* is the distance between nodes (0.25 mm); and *c*_A0_ is the phase velocity of the A_0_ mode at 43 kHz.

Three cases were used in the simulation study: defect-free, delamination between the first and second layers, and delamination between the second and third layers. The B-scan images obtained as a result of modelling Lamb wave propagation in a multilayer defective WTB structure are presented in Figure 5a–c. The B-scan image obtained in a defect-free region is presented in Figure 5a, in a region where delamination is between the first and second layers (Figure 5b) and in a region where delamination is between the second and third layers (Figure 5c). The filtering was performed as it is presented in Figure 2, in order to highlight the A_0_ mode. The obtained filtered B-scan images of all cases of the study are presented in Figure 5d,e,f. According to the proposed algorithm, in Figure 2, the phase velocity of the A_0_ mode is calculated for each case. The obtained phase velocity values in the defect free region, in the region with delamination between first and second layers, and in the region with delamination between the second and third layers are presented in Figure 6a,b,c, respectively.

The results obtained clearly show that in both simulated delamination cases, in the range from ~170 mm to ~250 mm, the phase velocity was reduced to 600–700 m/s. The coordinates of the first and second edges of the delamination in both cases were the same and equal to *x*_1_ = *x*_3_ = 172.5 mm and *x*_2_ = *x*_4_ = 255 mm, respectively, and the delamination size was determined to be *dl*_1_ = *dl*_2_ = 82.5 mm.

However, due to the interference phenomena, in the other mode seen in B-scan images (Figure 5d–f), scattered results were obtained in all cases of the study (Figure 6a–c). Thus, according to the calculated phase velocity values (Figure 6b,c), it is difficult to assess at which depth the delamination was and to determine between which layers it was present. Therefore, in order to obtain more accurate values, it was proposed to calculate the segments of the phase velocity dispersion curve. The zero-crossing method was applied for the calculation of the phase velocity dispersion curve segment of the A_0_ mode [35]. The necessary parameters that needed to be selected in each case of the study were two adjusted signals, the threshold level, and the number of time instances where the signals crossed the zero-amplitude line. Two signals at distances *x*_1_ = 60 and *x*_2_ = 200 mm were selected in a defect-free sample and the threshold level *U_L_* = 0.1 was set; *x*_1_ = 177 and *x*_2_ = 202 mm in the region with delamination between the first and second layers were selected and the threshold level *U_L_* = 0.15 was set, and *x*_1_ = 175 and *x*_2_ = 205 mm were selected in the region with delamination between the second and third layers, and the threshold level *U_L_* = 0.2 was set. The eight time instances were recorded in each case of the study. The obtained segments of the reconstructed phase velocity dispersion curves of the A_0_ mode in a defect-free region, in a region with delamination between the first and second layers, and in a region with delamination between second and third layers are presented in Figure 7.

The resulting segment of the phase velocity dispersion curve of A_0_ mode in the defect-free region was 1230–1300 m/s. This is consistent with the SAFE calculated dispersion curve presented in our previous work [34]. The segment of the phase velocity dispersion curve in the region with delamination between the first and second layers was obtained as 628–644 m/s, and in the region with delamination between the second and third layers, it was 685–708 m/s. The difference in phase velocity was about 60 m/s. The results clearly show that the phase velocity of the reconstructed dispersion curves varied with the delamination depth of the layers. Thus, the phase velocity dispersion curve can be used to determine the depth of delamination. Based on the simulation results, the proposed new algorithm is suitable for the detailed determination of the delamination parameters of a multilayer structure.

Since it was not possible to carry out the experiment with different types of delaminations in a WTB, a milled hole-type defect with a diameter of 81 mm was chosen for the study. It was assumed that a drilled hole, which has analogous characteristics with respect to Lamb wave propagation, can be used as a defect corresponding to delamination. This assumption was tested with additional theoretical simulations. The simulation results are shown in Figure 8. Delamination between the second and third layers (Figure 8a) and a hole drilled at the corresponding depth (Figure 8b) were selected for simulation. The out-of-plane components were analyzed in the simulation. The obtained B-scan images are presented in Figure 8c,d. The obtain phase velocity variation in the A_0_ mode at the defect location is presented in Figure 8e.

According to the algorithm proposed in Figure 2, the phase velocity variation in the A_0_ mode at the defect location was calculated for each case. The obtained simulation results (Figure 8e) confirmed the assumption that a drilled hole can be used as a defect corresponding to delamination. Thus, the experiment verification was performed using a milled hole-type defect.

## 3. Experimental Verification

The performance of the proposed new algorithm was experimentally verified on the WTB sample. A milled hole-type defect with a diameter of 81 mm was used, which had analogous characteristics with respect to Lamb wave propagation. The real WTB sample is presented in Figure 9.

The low-frequency ultrasonic system developed by the Ultrasound Research Institute of Kaunas University of Technology [36] was used for the experimental study. The macro-fibre composite (MFC) transducer of a P1-type MFC-P1-2814 (S.n. 08J10079l) transducer manufactured by Smart Materials was used for the excitation of ultrasonic guided waves. The characteristics of this MFC transducer have been studied in detail [37]. The wideband contact-type piezoceramic transducer was used as a receiver. The experimental setup of WTB inspection is presented in Figure 10a. In previous work [34], the resonant frequency of the MFC transducer was found to be 43 kHz, and therefore, this frequency was chosen as the excitation frequency. The transducer was excited by a rectangular pulse with a duration of 11.6 µs. The wideband contact-type ultrasonic receiving transducer was scanned up to the distance of *x*_2_ = 160 mm away from the transmitter with a step of 0.1 mm. The initial distance between the transmitter and receiver was *x*_1_ = 30 mm. The experimentally obtained B-scan images as a result of Lamb wave propagation in a multilayer defective WTB structure are presented in Figure 10b. According to the proposed algorithm, filtering was performed. The obtained filtered B-scan image is presented in Figure 10c.

According to the algorithm presented in Figure 2, the phase velocity was obtained in the defect-free and defect regions (Figure 11). The obtained phase velocity variation clearly indicated the defect region (Figure 11a). According to the calculated threshold level between the minimum and maximum values of the phase velocity, the size of the defect was determined, which was *dl*_1_ = 83.2 mm. The phase velocity of the A_0_ mode over the defect-free region was obtained as about 1260–1300 m/s, and over the defect region, it was about 780–810 m/s (Figure 11b). Two signals at distances *x*_1_ = 45 mm and *x*_2_ = 55 mm were selected in a defect-free sample and *x*_1_ = 90 mm and *x*_2_ = 110 mm in the defect region. The threshold level *U_L_* = 0.1 was set and six time instances were recorded in each case of the study. Applying the zero-crossing method, the segments of the phase velocity dispersion curves in the defect-free and defect regions were calculated (Figure 11b).

Based on the obtained results, the location, size, and depth of the defect could be evaluated using the proposed algorithm. The results obtained with experimental signals indicated that the proposed signal-processing algorithm is a suitable tool for the detailed determination of delamination parameters in a multilayer structure.

## 4. Conclusions

A new signal-processing algorithm is proposed for the detailed determination of delamination parameters of multilayer structures using the A_0_ mode of Lamb waves. The algorithm consists of two main parts: the calculation of the phase velocity variation and estimation of the dispersion of this velocity. Applying the first part of the proposed algorithm, the coordinates of the delamination edge are determined, which indicate the size of delamination. Applying the second part of the algorithm, information is obtained regarding at what depth or between which layers the delamination is. The simulation and experimental studies were conducted to validate the proposed technique. The WTB sample was chosen and the A_0_-mode signals at 43 kHz range were used in both studies. The proposed method was verified by the use of an 81 mm diameter WTB segment delamination in the simulation study. Four cases were used: defect-free, delamination between the first and second layers, delamination between the second and third layers, and defect (hole). The obtained phase velocity values showed the location and width of both delaminations. The delamination size in both cases was determined to be *dl*_1_ = *dl*_2_ = 82.5 mm. The reconstructed phase velocity dispersion curve of the A_0_ mode was obtained at 628–644 m/s in the region with delamination between the first and second layers and in the region with delamination between the second and third layers was obtained at 685–708 m/s. A simulation study was carried out to confirm that a drilled hole can be used as a defect corresponding to delamination. Then, the experimental study was carried out using the WTB sample with a drilled hole of 81 mm to validate the proposed algorithm. The phase velocity variations in the A_0_ mode, calculated according to the proposed algorithm, showed a defect size of 83.2 mm. The calculated A_0_-mode dispersion curve segments were obtained for the defect-free region at about 1260–1300 m/s and for the defect region at about 780–810 m/s. The simulated and experimental results indicated that the proposed signal-processing algorithm is a suitable tool for the parameterization of delamination in a multilayer structure.

## Figures and Tables

**Figure 1 sensors-25-00539-f001:**
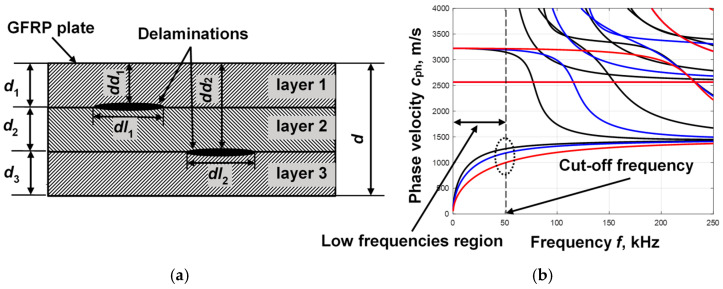
Three-layer GFRP with delaminations between the different layers (**a**) and phase velocity dispersion curves in defect-free and defective regions calculated by the SAFE method (**b**); black lines are dispersion curves in the defect-free region, and red and blue lines are in the first and second regions of delaminations.

**Figure 2 sensors-25-00539-f002:**
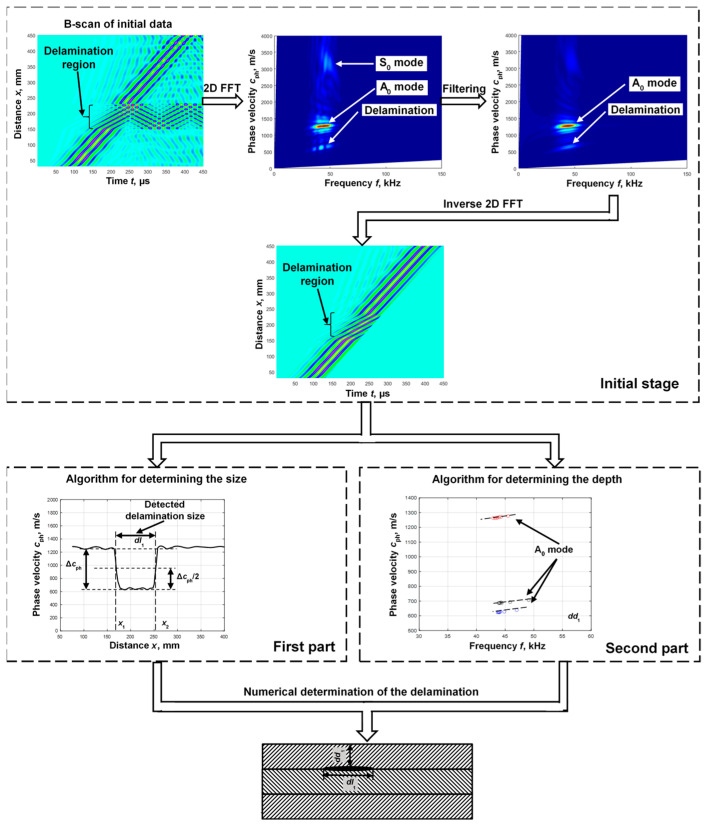
The flow chart of the proposed signal-processing algorithm for delamination parametrization.

**Figure 3 sensors-25-00539-f003:**
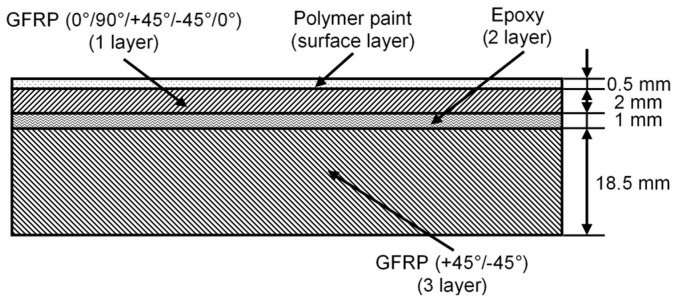
The graphical representation of the multilayer structure of a WTB.

**Figure 4 sensors-25-00539-f004:**
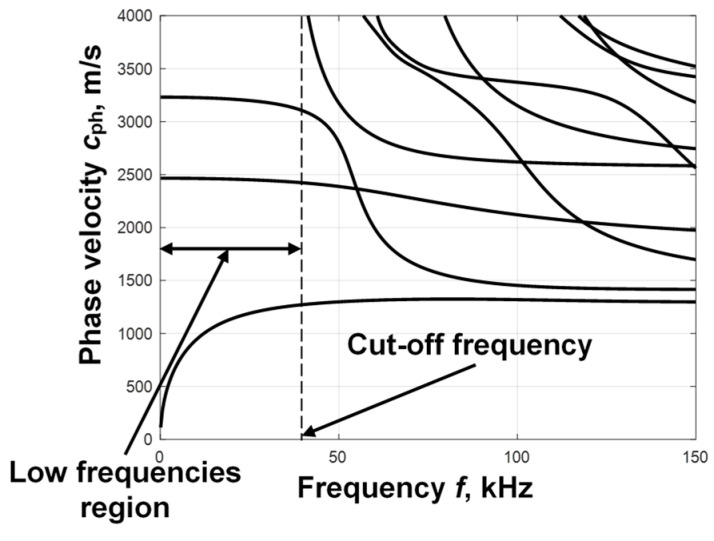
The phase velocity dispersion curves of the A_0_ and S_0_ modes calculated by the SAFE method in the WTB multilayer structure.

**Figure 5 sensors-25-00539-f005:**
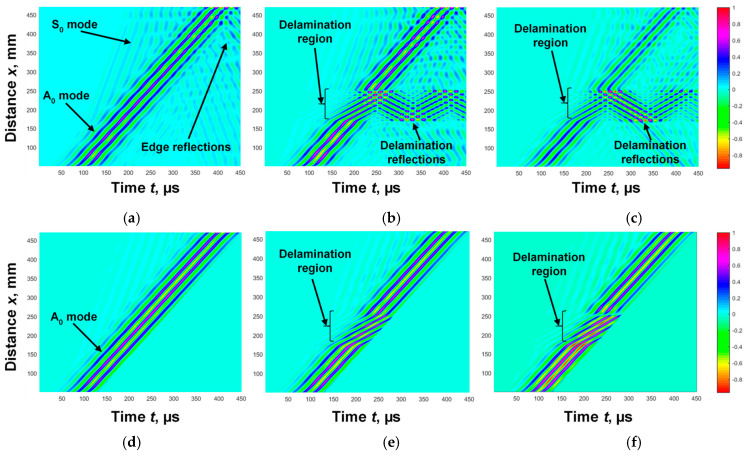
Simulated B-scan images of propagated Lamb wave in defect-free region (**a**), in region with delamination between 1st and 2nd layers (**b**), and in region with delamination between 2nd and 3rd layers (**c**); filtered B-scan images (**d**–**f**).

**Figure 6 sensors-25-00539-f006:**
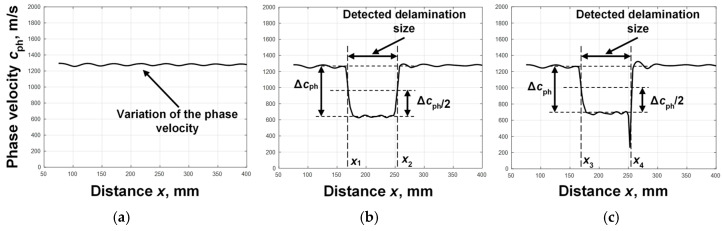
Variation in the phase velocity of the A_0_ mode of the Lamb wave with respect to the distance when there is no defect in the multilayer structure (**a**), when delamination is between the 1st and 2nd layers (**b**), and when delamination is between the 2nd and 3rd layers (**c**).

**Figure 7 sensors-25-00539-f007:**
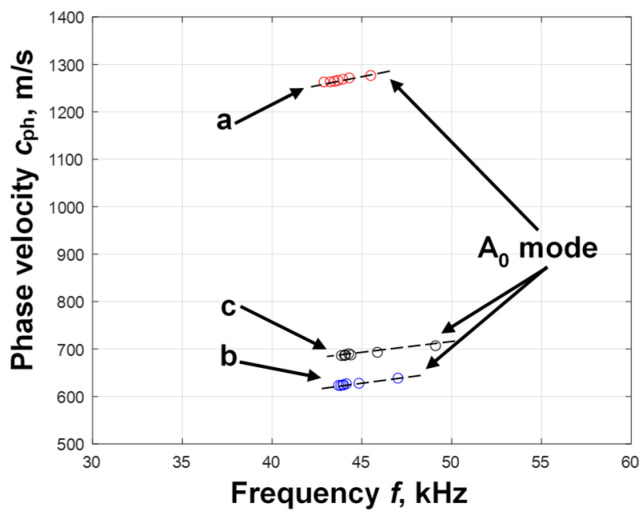
Variation in the phase velocity of the Lamb wave A_0_ mode with respect to the frequency when there is no defect in the multilayer structure (**a**), when delamination is between the 1st and 2nd layers (**b**), and when delamination is between the 2nd and 3rd layers (**c**).

**Figure 8 sensors-25-00539-f008:**
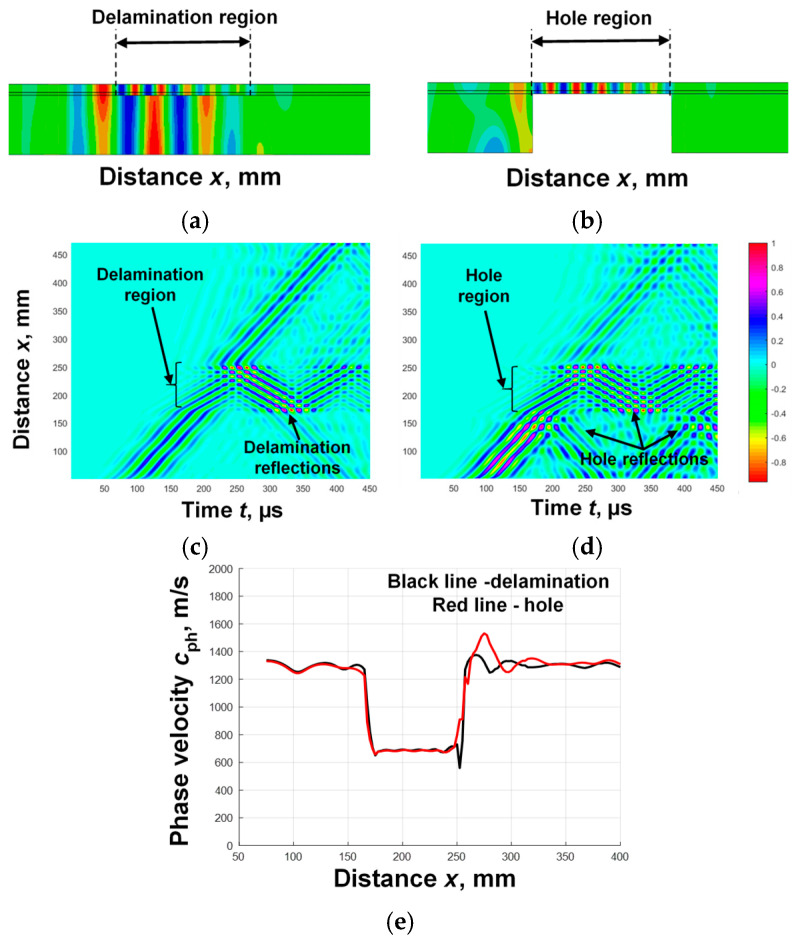
Simulation results of Lamb wave propagation in a WTB specimen with a delamination between the 2nd and 3rd layers (**a**), a hole drilled at the corresponding depth (**b**); the B-scan images are presented in (**c**,**d**) accordingly, and phase velocity variation in the A_0_ mode at the defect location is shown in (**e**).

**Figure 9 sensors-25-00539-f009:**
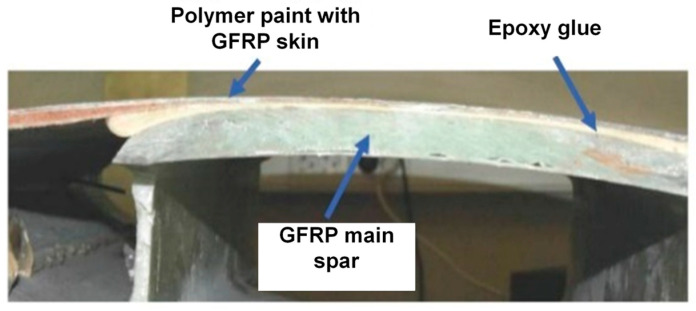
The real WTB sample.

**Figure 10 sensors-25-00539-f010:**
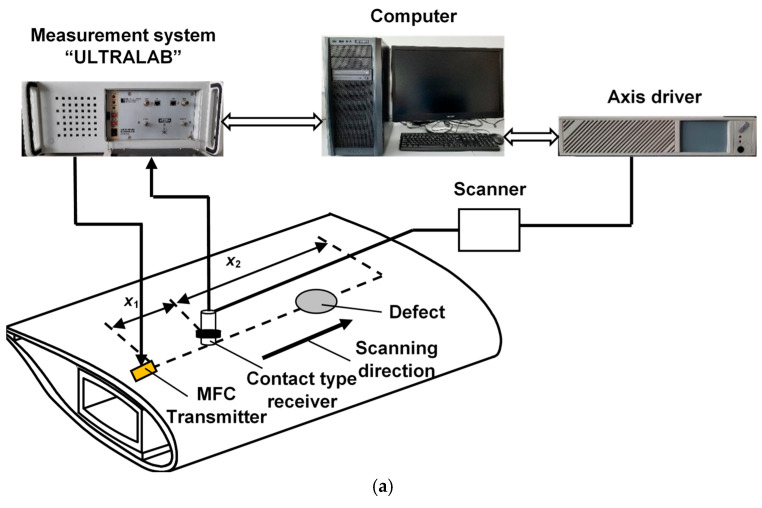
Experimental setup of WTB inspection (**a**), B-scan image of propagated Lamb wave in region (**b**) and filtered experimental B-scan image (**c**).

**Figure 11 sensors-25-00539-f011:**
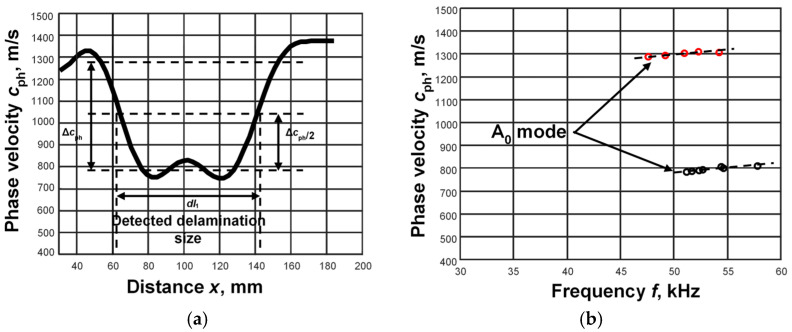
Variation in the phase velocity of the Lamb wave A_0_ mode in defect-free and defect regions (**a**); the segments of the phase velocity dispersion curves in defect-free and defect regions (**b**).

**Table 1 sensors-25-00539-t001:** Elastic parameters of WTB composite structure layer materials.

Parameters	Numerical Value
Paint (Surface layer):	
Density (*ρ*)	1270 kg/m^3^
Young’s modulus (E)	4.2 GPa
Poisson’s ratio (υ)	0.35
Unidirectional GFRP layer:	
Density (*ρ*)	1828 kg/m^3^
Young’s modulus (E_1_)	42.5 GPa
Young’s modulus (E_2_)	10 GPa
Poisson’s ratio (υ_12_)	0.26
Poisson’s ratio (υ_23_)	0.4
In-plane shear modulus (G_12_)	4.3 GPa
Epoxy:	
Density (*ρ*)	1260 kg/m^3^
Young’s modulus (E)	3.6 GPa
Poisson’s ratio (υ)	0.35

## Data Availability

Data are contained within the article.

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
