# Peer review of "Detailed Determination of Delamination Parameters in a Multilayer Structure Using Asymmetric Lamb Wave Mode"

_sensors, 2025, doi:10.3390/s25020539_

Round 1

Reviewer 1 Report

Comments and Suggestions for Authors

Comments:

This study introduces a novel signal-processing algorithm for precisely determining delamination parameters in multilayer composite structures, utilizing the phase velocity characteristics of the Lamb wave A0 mode. The algorithm incorporates phase velocity dispersion analysis to accurately identify the size, location, and depth of delamination. Validation was performed through finite element simulations and experimental wind turbine blade sample investigations. The results demonstrate a strong correlation between phase velocity variations and delamination depth, confirming the algorithm's effectiveness in addressing critical challenges associated with the non-destructive evaluation of complex multilayered structures. However, the following issues are suggested to be well addressed to improve the manuscript quality.

1.      Would including a comparative analysis of experimental data further substantiate the effectiveness and superiority of the proposed method?

2.      In the simulation analysis, did the authors consider the actual boundary conditions and constraints of the wind turbine blade (WTB) structure, or was a simplified flat plate model employed? Since the WTB is inherently a curved structure, has the consistency between the simulation results and the experimental data been rigorously evaluated and demonstrated?

3.      In the experimental section, does the drilled hole adequately represent delamination damage? As a through-thickness defect, a drilled hole fundamentally differs from delamination in terms of its characteristics and the mechanisms it affects wave propagation. Could this suggest that the proposed method is not exclusively designed for delamination detection but also applies to identifying other types of structural defects?

4.      This study introduces a method for determining the depth of delamination. Would it be advisable to conduct experiments with delaminations at varying depths to rigorously evaluate and further validate the accuracy and effectiveness of the proposed method?

5.      Why choose 43 kHz frequency?

6.      Kindly revise the abstract and introduction sections to enhance the clarity and precision of the research findings, ensuring that they are articulated with greater specificity to emphasize the study's focus while distinctly highlighting the methodological innovations presented in the work.

7.      What is the scientific rationale for utilizing MFC to excite ultrasonic guided waves? What are the specific physical characteristics and structural details of the MFC? The lack of a sensor illustration in the manuscript raises concerns about whether this omission impacts the comprehensiveness of the methodological description.

Author Response

  1. Would including a comparative analysis of experimental data further substantiate the effectiveness and superiority of the proposed method?

The aim of this article was to present a proposed model for estimating the size of delamination and to validate it experimentally. Of course, a comparative analysis of the experimental data would provide more evidence of the effectiveness of the method. Our future work will focus on investigating the effectiveness of the method for different defect sizes and thicknesses of the object under the investigation.

  1. In the simulation analysis, did the authors consider the actual boundary conditions and constraints of the wind turbine blade (WTB) structure, or was a simplified flat plate model employed? Since the WTB is inherently a curved structure, has the consistency between the simulation results and the experimental data been rigorously evaluated and demonstrated?

In the numerical modelling simplified 2D plane strain model was used. Main task of the modelling was investigation of guided waves interaction with delamination type defects at different depths and simplified plate model is enough to solve such task. On the other hand, measurements on wind turbine blade were performed along relatively plane area where main spar is bonded with composite skin and curvature of WTB can be neglected (see Figure 10,a). In our opinion, curvature with such big radius will not affect propagation of Lamb waves.

  1. In the experimental section, does the drilled hole adequately represent delamination damage? As a through-thickness defect, a drilled hole fundamentally differs from delamination in terms of its characteristics and the mechanisms it affects wave propagation. Could this suggest that the proposed method is not exclusively designed for delamination detection but also applies to identifying other types of structural defects?

In order to compare the characteristics of Lamb wave propagation in the presence of delamination and drilled hole defects, the text of the article is expanded with the results of theoretical simulations (Figure 8). The phase velocity values were compared with the delamination of the 2nd and 3rd layers and a hole drilled at the corresponding depth. The obtained simulation results confirmed that the drilled hole can be used as a defect corresponding to delamination. However, the proposed method cannot be fully used for other types of defect detection. It can be used to determine the width of the drilled hole, but it cannot determine how deep it was drilled.

  1. This study introduces a method for determining the depth of delamination. Would it be advisable to conduct experiments with delaminations at varying depths to rigorously evaluate and further validate the accuracy and effectiveness of the proposed method?

In this phase of the study, we wanted to test the suitability of the method for the determination of structural defects and delamination. We obtained good results. A comparative analysis of the experimental data would therefore provide more evidence of the effectiveness of the method, and in future work we will focus on determining the effectiveness of the method for different depths of delamination.

  1. Why choose 43 kHz frequency?

The MFC-P1-2814 (S.n. 08J10079l) transducer manufactured by Smart Materials was used for experimental research. The characteristics of this MFC transducer have been studied in detail [1]. According to research findings, the most optimal excitation frequency is around 43 kHz. This excitation frequency met the requirement of being lower than the cut-off frequency of the higher modes (Figure 4).

Tiwari, K.A.; Raisutis, R. Investigation of the 3D displacement characteristics for a macro-fiber composite transducer (MFC-P1). Mater. Tehnol. 2018, 52, 235–239.

  1. Kindly revise the abstract and introduction sections to enhance the clarity and precision of the research findings, ensuring that they are articulated with greater specificity to emphasize the study's focus while distinctly highlighting the methodological innovations presented in the work.

The abstract and introduction sections were revised. Additional information was added. The introduction section has been expanded and extended.

  1. What is the scientific rationale for utilizing MFC to excite ultrasonic guided waves? What are the specific physical characteristics and structural details of the MFC? The lack of a sensor illustration in the manuscript raises concerns about whether this omission impacts the comprehensiveness of the methodological description.

The P1-type macro-fiber composite transducers (MFC) are widely used as Lamb wave transmitters for NDT and SHM of composite structures [2]. These transducers are easy to embed in long and complex structures due to their small size, lightweight, high reliability and durability.

William, K.W.; Bryant, R.G.; High, J.W.; Robert, L.F.; Hellbaum, R.F.; Jalink, A.; Little, B.D.; Mirick, P.H. Low-cost piezocomposite actuator for structural control applications. SPIE Proc. 2000.

Reviewer 2 Report

Comments and Suggestions for Authors

The paper, "Detailed Determination of Delamination Parameters in a Multi-Layer Structure Using Asymmetric Lamb Wave Mode," presents a novel and interesting approach to detecting and characterizing delamination in multilayer composite structures using Lamb waves. The research is highly relevant to the fields of nondestructive testing (NDT) and structural health monitoring (SHM), particularly in applications like wind turbine blades, where early detection of damage is critical. The authors’ proposed signal-processing algorithm is innovative and shows great potential, supported by simulations and experimental work. However, there are areas where the paper could be improved to better communicate its findings and make the study more impactful.

The introduction does a good job of framing the problem and explaining why delamination detection is important, though adding a few more recent references could strengthen the foundation. The methodology is detailed and provides a clear explanation of the algorithm and experimental setup. The results section is thorough in its simulations, but the experimental results feel somewhat limited. Including more experimental data, along with some statistical analysis or repeatability studies, would make the findings much more robust. The proposed experimental work does not align well with the simulations. A delamination affects the Lamb wave propagation in a totally different manner than a milled hole. If it is not possible to run an experiment with a structure similar to the simulated one, then it is recommended that the simulation also include a milled hole to study its effect on Lamb waves and compare it with the delamination.

In summary, this is a creditable piece of work with a lot of potential, but it needs some revisions to reach its full impact. Providing more experimental validation and refining the discussion. I recommend minor revisions before publication, but I am confident that these changes will make the paper even stronger and more impactful.

Author Response

The introduction does a good job of framing the problem and explaining why delamination detection is important, though adding a few more recent references could strengthen the foundation. The methodology is detailed and provides a clear explanation of the algorithm and experimental setup. The results section is thorough in its simulations, but the experimental results feel somewhat limited. Including more experimental data, along with some statistical analysis or repeatability studies, would make the findings much more robust. The proposed experimental work does not align well with the simulations. A delamination affects the Lamb wave propagation in a totally different manner than a milled hole. If it is not possible to run an experiment with a structure similar to the simulated one, then it is recommended that the simulation also include a milled hole to study its effect on Lamb waves and compare it with the delamination.

In order to compare the characteristics of Lamb wave propagation in the presence of delamination and drilled hole defects, the text of the article is expanded with the results of theoretical simulations (Figure 8).

Reviewer 3 Report

Comments and Suggestions for Authors

The paper discusses the development of a method for inspecting the health of composite structures. It builds on previous work, so it's important to clearly show what is new compared to previously published studies.

Please check carefully the correspondence between references used and the text in the Introduction. It would also be helpful to expand on the state-of-the-art in acoustic techniques for delamination detection.

Reference 10 refers to wind turbine monitoring, so this should be mentioned in the Introduction initially.

Have you considered the dependence of group velocity on delamination? How were the red and blue lines obtained in Figure 1b?

How were the red and blue lines in Figure 1b obtained?

The Ref.16 and Ref 23 are the same.

What is the main difference between this paper and the paper published earlier by the same authors' group? What are the "new" (lines 270 and 318) of the proposed algorithm compared to those presented in [16] or [23]?

There is no ref.[37] (line 283) in the Literature.

Comments on the Quality of English Language

English needs to be checked and proofread.

Author Response

Please check carefully the correspondence between references used and the text in the Introduction. It would also be helpful to expand on the state-of-the-art in acoustic techniques for delamination detection.

The introduction section has been expanded and extended. References checked and inconsistencies corrected.

Reference 10 refers to wind turbine monitoring, so this should be mentioned in the Introduction initially.

It is done.

Have you considered the dependence of group velocity on delamination?

Phase velocity measurement allows you to accurately measure the velocity using two adjacent signals. In this way, the location of the velocity change can be accurately measured during scanning and the zone of defect can be identified. Of course, there will be a variation in the group velocity in the defected zone. However, the measurement of the group velocity between adjacent signals leads to large errors. Measurement of the group velocity over large distances between transducers can be accurate, but the exact location and dimensions of the defect cannot be determined. The measurement will only indicate that there is a change in the object under test compared to a known non-defective location.

How were the red and blue lines in Figure 1b obtained?

At the place of delamination, the structure under investigation becomes thinner in relation to the propagation of Lamb waves and the propagation of these waves changes. The thinning of the structure depends on the layers between which there is delamination. Therefore, different dispersion curves of Lamb wave propagation are obtained for different delamination locations, which are determined by the thickness of the structure. Such dispersion curves are depicted in Fig. 1. The blue line represents the dispersion curve for delamination between the first and second layers, and the red line for the delamination between the second and third layers.

The Ref.16 and Ref 23 are the same.

This has been corrected.

What is the main difference between this paper and the paper published earlier by the same authors' group? What are the "new" (lines 270 and 318) of the proposed algorithm compared to those presented in [16] or [23]?

The algorithm presented in the previously published article enables to determine the location of the defect and its size with respect to only one coordinate (length). Meanwhile, the proposed new improved algorithm expands the possibilities of use in relation to another coordinate (thickness). Unfortunately, in this case, the proposed algorithm is applicable only to the determination of the depth of the delamination defect.

There is no ref.[37] (line 283) in the Literature.

This has been corrected.

Round 2

Reviewer 1 Report

Comments and Suggestions for Authors

The author answered all questions.

Reviewer 3 Report

Comments and Suggestions for Authors

Thank you for replies and corrections